# An Update on In Vitro Folliculogenesis: A New Technique for Post-Cancer Fertility

**DOI:** 10.3390/biomedicines10092217

**Published:** 2022-09-07

**Authors:** Elsa Labrune, Bruno Salle, Jacqueline Lornage

**Affiliations:** 1Hospices Civils de Lyon, Hôpital Femme Mère Enfant, Service de Médecine de la Reproduction, 59 boulevard Pinel, 69500 Bron, France; 2Faculté de Médecine Laennec, Université Claude Bernard, 7 rue Guillaume Paradin, 69008 Lyon, France; 3INSERM Unité 1208, 18 avenue Doyen Lépine, 69500 Bron, France; 4Faculté de Médecine Lyon Sud, Université Claude Bernard, 165 Chemin du Petit Revoyet, 69600 Oullins, France

**Keywords:** oncology nursing, folliculogenesis, culture medium, oogenesis, assisted reproductive technology, fertility preservation

## Abstract

Introduction: Obtaining in vitro mature oocytes from ovarian tissue to preserve women’s fertility is still a challenge. At present, there is a therapeutic deadlock for girls and women who need emergency fertility preservation in case of a high risk of ovary invasion by malignant cells. In such a case, ovarian tissue cannot be engrafted; an alternative could be in vitro folliculogenesis. Methods: This review focuses on the progress of in vitro folliculogenesis in humans. PubMed and Embase databases were used to search for original English-language articles. Results: The first phase of in vitro folliculogenesis is carried out in the original ovarian tissue. The addition of one (or more) initiation activator(s) is not essential but allows better yields and the use of a 3D culture system at this stage provides no added value. The second stage requires a mechanical and/or enzymatic isolation of the secondary follicles. The use of an activator and/or a 3D culture system is then necessary. Conclusion: The current results are promising but there is still a long way to go. Obtaining live births in large animals is an essential step in validating this in vitro folliculogenesis technique.

## 1. Introduction

Obtaining in vitro mature oocytes from ovarian tissue is a challenge to understanding oogenesis and folliculogenesis phenomena as well as preserving female fertility. For the latter purpose, cryopreservation of ovarian tissue remains the only technique likely to be proposed to prepubertal girls and women whose potentially gonadotoxic treatment (chemotherapy, radiotherapy) is urgent or to those who are not compatible with ovarian hyperstimulation. The use of cryopreserved ovarian tissue requires transplantation when the woman wishes to become pregnant [1]. Since the live birth reported in 2004 by Donnez et al. [2], the results have been satisfactory; the success rate regarding recovery of endocrine activity amounts to more than 90% and that regarding live births to more than 30% [3,4]; however, autografts were not recommended when there was a risk of reintroduction of malignant cells, as in acute leukemia [5] or borderline ovarian tumors [6]. An interesting alternative to ovarian tissue transplantation can be in vitro folliculogenesis. In case of a parental project, the ovarian tissue can be thawed, and then cultured for in vitro folliculogenesis enables production of mature oocytes destined for in vitro fertilization [1].

Ovarian tissue contains mostly primordial and primary follicles, which constitute a reserve pool; it also contains small follicles that have a high level of resistance to gonadotoxic treatments and freezing techniques [7,8]. Performing a complete in vitro folliculogenesis consists then in initiating follicular growth from this reserve pool so that to obtain antral follicles that include a cumulo-oocyte complex (COC). The COC may be matured in vitro until it becomes a mature oocyte in metaphase II, then used for in vitro fertilization. The first live birth resulting from complete in vitro folliculogenesis was obtained in mice in 1996 by Eppig et al. [9] and repeated in 2003 [10]. In large animals, the results were encouraging too; the first step was the activation of primordial follicles that allowed growth to the secondary follicle stage; a process that was reported by several teams [11,12,13,14]. The second step consisted in cultures of secondary follicles until the maturation of oocytes and embryos; this was also reported in various species: the rat [15], the pig [16]; the buffalo [17], the sheep [18], the goat [19], and rhesus macaque [20]). Recently, the first pregnancy in goat from in vitro secondary follicle culture was reported [21]; however, no team reported having obtained mature oocytes from primordial follicles. The amount and quality of mature oocytes and embryos remained low: to the best of our knowledge, there is yet no report on large animal birth. In humans, studies are limited due to the difficult access to ovarian tissue and the technical difficulties and length (in months) of this tissue culture [22]. Indeed, the main challenge of in vitro folliculogenesis is due this long culture period because gas exchanges and nutrient, hormone, and growth factor supplies depend on the culture duration; however, the results obtained over the last 20 years do inspire optimism; e.g., the successful maturation of oocytes from the pre-antral stage in 2015 [23] and from the unilaminar follicle in 2018 and 2021 [24,25]. 

The present review intends to summarize the progress of in vitro folliculogenesis in humans; it focuses on the culture media and then, according to the culture stage, on the different culture systems developed with comments on the results obtained.

## 2. Methods

This systematic review was conducted according to the Preferred Reporting Items for Systematic Reviews and Meta-analysis (PRISMA) guidelines [26].

PubMed and Embase databases were searched for articles of interest published up to March 2022. The selection included all original English-language articles on in vitro folliculogenesis from ovarian tissue in humans that were returned with the following keywords and operators: (in vitro growth OR folliculogenesis) AND (follicle OR oocyte) AND (humans OR Hum Reprod [Journal] OR women OR woman). 

The selection excluded non-English papers, works on animals, in vitro maturation and in vivo maturation works carried out within the context of in vitro fertilization protocols, studies on in vitro folliculogenesis that checked slow freezing and/or vitrification of ovarian tissue, studies on frozen or vitrified tissues (these do not have the same objective), studies on short culture times, and studies that lacked major results. 

This search, inclusion, and exclusion strategy identified first 5463 articles. After the removal of duplicates and abstract review, 127 articles were kept for further reading. Finally, 24 articles underwent in-depth analyses and detailed culture conditions (Figure 1). 

This number might seem low probably because of too restrictive search criteria regarding the language (English only) and the species (human only). Nevertheless, the technique is relatively recent and there are currently very few teams dedicated to the search in that difficult domain.

The manuscript reports only on the most successful searches in the most popular and/or reliable databases. Searches in other databases (Web of science, Google Scholar, etc.) returned either duplicates or articles far from the scope or inclusion criteria of this literature review (Figure 2).

## 3. Description of Culture Media Used for In Vitro Folliculogenesis

The two most frequently used culture media are alphaMEM [7,23,25,27,28,29,30,31,32,33,34] and McCoy’s 5a [24,35,36,37,38,39] which is supplemented respectively with bicarbonates at 2.2 g/L or HEPES between 20 mM [36,37] and 25 mM [24,35]. The alphaMEM medium is more concentrated in amino acids than McCoy’s 5a medium; it is also more concentrated in ascorbic acid, an antioxidant that promotes follicular vitality during long in vitro culture [40] and in inorganic salts such as calcium chloride and sodium chloride. Inversely, McCoy’s 5a medium is more concentrated in vitamins such as biotin, folic acid, myoinositol as well as in glucose, lipoic acid, and sodium pyruvate. The results of studies using these culture media are comparable; there was no superiority of either medium over the other; however, it is important to know their compositions in order to supplement according to each culture’s needs. The main supplements found were glucose for alphaMEM and ascorbic acid for McCoy’s 5a. 

Other media were less used and their results were less advantageous in terms of diameter (secondary follicle) and follicular survival. A comparison between Earle’s balanced salt solution, alphaMEM, and Waymouth’s Medium found a superiority of alphaMEM in terms of follicular growth (increase in the diameters after 10 days of culture) and increase in the number of preantral follicles after 15 days of culture [41]. The media were supplemented with human serum albumin or previously desensitized patient serum. 

Most studies reported also supplementation with follicle stimulating hormone (**FSH**) [7,24,25,27,31,32,33,42,43,44], LH [24,42,45], insulin, transferrin, selenium (ITS; [7,23,24,25,29,30,31,32,33,34,36,37,39,46]), glutamine [24,35,36,37,39,44], and ascorbic acid [7,24,32,34,35,37,39,47] (Table 1).

Glucose supplementation should also be discussed when composing the medium. Too high glucose-concentrated medium might be harmful to the embryos with an alteration of mitochondrial functions. A study carried out on equine animals reports better mitochondrial function when the glucose concentration is physiological (5.6 mM). A supraphysiological concentration of 17 mM feeds through to the decrease in the glycolytic index and in the decrease of ATP-coupled respiration in the COCs [54]. The glucose concentration in the alphaMEM medium is 5 mM, whereas it is 17 mM in McCoy’s 5a medium. Results on human in vitro folliculogenesis do not allow us to conclude significantly on the effect of glucose concentration. 

According to these data, the medium with promising results is alphaMEM; it gave equivalent results to McCoy’s 5a and has the advantages of being supplemented with albumin and having lower glucose concentration and low concentration in FSH, insulin (in the ng/mL range), transferrin, selenium, and ascorbic acid (all in the µg/mL range).

## 4. From the Reserve Follicle to the Secondary Follicle

### 4.1. Tissue or Isolated Follicle Culture

The growth of reserve follicles is gonadotropin-independent and is a complex phenomenon that is still partly obscure; this step has been carried out in vitro in humans to allow the growth of primordial follicles into pre-antral follicles and was most often performed in situ [7,24,25,27,29,31,35,37,38,43,55]. When the primordial follicles were initially isolated, it was difficult to make them grow; these follicles increased in size during the first 24 h [44] and then underwent atresia at the third [7] or the seventh day [33]. Only two teams succeeded in cultivating isolated primordial follicles and seeing them evolve, for the most part, to the secondary follicle stage after 8 days of culture [32] or after 10 days of culture [34]. The tissues used and described in this review were fresh (no cryoconservation). Tissues cultured in vitro were small in size; 1 × 1 mm [7] to 4 × 2 mm [35] and very thin too; 0.3 mm [43] to 1–2 mm [31]. Dissecting the ovarian tissue to obtain ultra-thin samples was is carried out either with a magnifying glass, a scalpel, and needles for mechanical dilaceration [43] or with a Thomas Stadie-Riggs Tissue Slicer [7]. The published data were in favor of in situ culture at this stage of follicular development, small size tissues, and, especially, the lowest possible tissue thickness; the best results were reported with thicknesses of 0.5 to 1 mm (Table 2). 

### 4.2. Culture Systems

There were different culture systems in two dimensions (2D) and three dimensions (3D). A 2D system allowed follicle adhesion to a surface: a culture box well, a protein matrix such as collagen, or Matrigel^®^ (a solubilized basement membrane preparation extracted from Engelbreth-Holm-Swarm mouse sarcoma; Corning, Corning, NY, USA). The follicles were then covered with culture medium. The 3D culture systems allowed keeping the 3D architecture of the follicles (e.g., by encapsulation) and the links between oocytes and granulosa cells. The use of a culture system varied in the literature from no culture system at all [24,25,35,36,37,38,55] to 2D culture system [29,43] and even to 3D system [7,31,32,34,41]. The histological quality of the pre-antral follicles obtained from the reserve follicles was similar regardless of the culture technique. In the absence of a culture system, the mean increase in the proportion of secondary follicles during culture was 21.1% [24,35,36,37,38,55]. The increase with 2D culture systems consisting of protein matrices was 31.4%. [27,29,43]. The increase with 3D culture systems was more difficult to interpret because authors usually grouped primary and secondary follicles into a single category, termed “growing follicles”; still, the increase in the proportion of growing follicles reached 68.7% [31,32,34] (Table 2). Here, it is interesting to note that the two teams that performed complete folliculogenesis in vitro used no culture system [24,25].

### 4.3. Supplementation of Culture Systems: Use of Activators

#### 4.3.1. Spontaneous Activation

Growth initiation occurs spontaneously in the absence of activators [38]; this was attributed to a lifting of the Hippo signaling pathway inhibition during ovaries dissection as dissection changes intercellular tensions [56]. The Hippo pathway is composed of a set of negative growth regulators that act on a kinase cascade that inactivates Yes-associated protein (YAP)/transcriptional coactivator with PDZ-binding motif (TAZ) effectors. YAP/TAZ are related to factors playing a role in cell contraction. When this pathway is interrupted (possibly by tension changes), YAP/TAZ activates and causes increases in cell growth factors and apoptosis inhibitors [57,58,59]. 

#### 4.3.2. Activators

Supplementation of culture media during initiation phase was used to improve the yield of growing follicles [29,31,32,35,36,39]. Activator addition increased the yield of secondary follicles by 13.9 ± 8.9% secondary follicles [24,37,38,55] and up to 35.6 ± 3.1% [35,36,39]. All studies agreed that activators increase in follicular growth. Examples or activators are: (i) activators of the PI3K/Akt/mTOR pathway: dipotassium bisperoxo (5-hydroxypyridine-2carboxyl)oxovanadate (V) (bpV) which is a phosphatase and tensin homologue inhibitor deleted on chromosome 10 (PTEN) [39,60], target of rapamycin (mTOR) [35]; (ii) growth factors: vascular endothelial growth factor A165 (VEGF A165) [36], platelet-rich plasma (PRP) [34], basic fibroblast growth factor (bFGF) [32]; or, (iii) growth differentiation factor 9 (GDF9) [29] (Table 3). 

Stimulation of the Akt (serine/threonine kinase) pathway was used to allow the growth of primordial follicles of ovarian tissue in women with premature ovarian failure. Grafting ovarian tissue previously exposed to an Akt stimulator resulted in live births [58,61]. The PI3K/Akt/mTOR pathway, once activated, results in massive growth of primordial follicles; it can be directly activated by mTOR or via inhibition of PTEN, an inhibitor of this signaling pathway. bpV(HOpic) is a PTEN inhibitor that allowed the growth of primordial follicles in mice [62,63] as well as the development of mature follicles and mature oocytes [49]. The two studies reporting on bpV activation over 12 and 6 days recorded 41% and 46.2% decrease in primordial follicles but a 46% and 60% increase in primary plus secondary follicles [31,39]. bpV was not subsequently used in humans. 

#### 4.3.3. Growth Factors

VEGF A165 was used to stimulate the proliferation, migration, and survival of endothelial cells; it induces the proliferation of spermatogonia [64] and is present in the granulosa and theca cells of human and bovine secondary follicles [65]. Supplementation with VEGF A165 allowed the development of bovine primary and secondary follicles [66]. The bFGF is also involved in follicular development. In humans, bFGF and its receptor are expressed at the initiation of reserve follicles; it also had a role in the production of estrogen by ovarian tissue [32,67]. 

PRP is a set of growth factors present in platelets: platelet-derived growth factor (PDGF), transforming growth factor beta (TGFβ), vascular endothelial growth factor (VEGF), epidermal growth factor (EGF), fibroblast growth factor (FGF), and insulin growth factor (IGF). Finally, GDF9 (a member of the TGFβ family) is secreted by the oocyte and expressed in early follicular development [29].

In humans, the growth from primordial follicle to pre-antral follicle is possible provided that the follicles are left within the tissue. The continuation of folliculogenesis from the pre-antral stage to the antrum has not been reported with a follicle still within the tissue. The two techniques that allowed continued folliculogenesis within the tissue are promising but not currently usable in humans: (i) xenotransplantation of human ovarian tissue into the mouse kidney capsule [49]; and, (ii) decellularization of human ovary but keeping only the extracellular matrix, isolation of the follicles and their insemination into the structure, then the ovary is xenografted into an immunodeficient mouse [68]. Two hypotheses would explain in situ culture failures from the pre-antral follicle stage: (i) the change in cell density around the follicle during folliculogenesis: high when the reserve follicle is in the ovarian cortex to low when the secondary follicle migrates to the medulla (note that the secondary follicle is kept in the cortex with no density change); (ii) the increase in nutrient and gas supply when the follicle migrates from the cortex to the medulla; these hypotheses are not mutually exclusive, which explains the need for isolating secondary follicles obtained in vitro. 

## 5. From the Secondary Follicle to the Antral Follicle

### 5.1. Isolation of Follicles

Isolating secondary follicles is delicate; it can be enzymatic, mechanical, or a succession of both; the latter being the most common [69] (Table 4). Careful isolation seem necessary for in vitro continuation of folliculogenesis. Enzymatic digestion did not alter follicles; these were able to grow in culture [69,70]; however, all three teams that obtained metaphase II oocytes used mechanical digestion only [23,24,25]. Most enzymatic protocols used one or more collagenases only or in combination with a protease (Liberase DH) or DNAse [70,71]. Quantitative and qualitative results in isolated follicles were quite similar on pubescent human tissue without exposure to gonadotoxic treatments or androgens [71]. The digestion protocol use should be adapted to the collagen fiber content of the ovarian tissue of interest (fetal, pre-pubescent, exposed to drugs).

### 5.2. Culture Systems

Once isolated from the original ovarian tissue, the follicles are grown without support [24,25], in a 2D culture system [28,42,44,46], or, more recently, in a 3D system [7,23,30,33,34,45,47] (Table 5). 

Unsupported culture was carried out by McLaughlin’s team and was of short duration; secondary follicles reached the antral follicle stage in 8 days [24]. More recently, another team carried out cultures without support over longer durations (9 weeks) to reach the antral stage [25] The other teams adopted 3D culture systems given the results of 2D cultures [72]. The follicles lost the three-dimensional architecture observed in vivo. The composition of these 3D systems is variable. The most common biomaterial in humans was alginate, a natural polymer derived from algae, that proved to be non-cytotoxic, permeable to nutrients and gas exchanges (depending on its concentration), and easy to use. Concentrations of 0.5 ± 0.2% [23,30,33,47] had a low impact on follicular development [7]. Alginate may be used as a follicle encapsulation bead or tissue formed with a 3D printer [73]. The longest culture times with alginate 3D systems ranged from 30 to 40 days [7,23,30,33]. 

Collagen and fibrinogen from living organisms were also used but only in animals, including mice [74]. Another culture system tested in humans was a reconstitution of a complex from granulosa and thecal cells from patients stimulated in IVF; it proved difficult to use in routine practice [45]. A less well-known biomaterial was chitosan; it was tested in mice and seemed promising [75]. Finally, among the most recent culture systems developed in humans was the above-described ‘decellularization’ technique (See Section 4.3.3) [68,76].

Support or 3D structure did not seem essential to in vitro culture of secondary follicles to reach the antrum follicle stage [24,25]; they allowed however longer growing periods [7,23,30,33] that tended to reach human physiological ones reported by Gougeon [22]. Long cultures may promote the development of functional oocytes that requires accumulation of proteins and organelles essential for fertilization and embryonic development [77].

### 5.3. Supplementation of the Culture Medium

Telfer’s team and Xu’s team grew secondary follicles without support but supplemented the culture medium with activators [24,25,38]; the other teams did not use activators at the secondary follicle stage [7,23,28,30,33,42,45,46,47] (Table 5). The main activator used from the secondary follicle stage onwards was activin A; it was used in 2008 by Telfer’s team who reported the formation of follicles in the antrum in 2018 as the production of meta-phase II oocytes [24,38]. Activins are dimeric glycoproteins of the TGFβ family; their two subunits linked by a disulfide bridge form inhibin. Activin A is not only a hormone, but also a growth factor and a cytokine; it is involved in reproductive physiology. At the oocyte level, the paracrine actions of activin A are integrated within multiple and complex signaling pathways that ultimately modulate transcription genes at the nuclear level [78]. In primary and secondary follicles, activin A is expressed by granulosa cells and its receptors and mediators (SMAD2, SMAD3, and SMAD4) expressed in the oocyte [79]. Pre-antral and antral follicles synthesize activin A that stimulates the proliferation of granulosa cells, the formation of an antral cavity, and follicular growth [80]. Within the antral follicle, activin A also stimulates steroidogenesis; it increased aromatase activity induced by FSH in rat, bovine, and primate antral follicles [81,82,83]. In addition, Activin A stimulates the expression of estrogen receptors in mice [84]. Finally, in mice, the oocyte resumption of meiosis occurred concomitantly with an increase in activin A [85].

Xu’s team modulated anti-Mullerian hormone (AMH) at various stages of follicular development through supplementation with AMH or anti-AMH antibody. AMH is produced by granulosa cells and influences folliculogenesis; its secretion starts with the growth of the ovarian follicle to a secondary follicle with small antrum but decreases as the follicles become sensitive to gonadotropins [86]. AMH was added at the passage from the secondary follicle stage without antrum to the secondary follicle stage with a small antrum, then an anti-AMH antibody was added [25].

## 6. In Vitro Maturation of Cumulus Oocyte Complexes and Oocyte Quality

This technique was proposed to women to reduce the dose of gonadotropins and limit the risk of ovarian hyperstimulation [87]; it consists in puncturing small (6–12 mm) antral follicles not stimulated by external gonadotropins. Recovered COCs are then placed in a maturation medium to obtain metaphase II oocytes ready for fertilization. In 1995, Zhang et al. were the first to use the technique from COCs in vitro cultures (Table 5). The tissues were fetal human tissues at 16 to 20 weeks gestation and a quarter of COCs contained metaphase II oocytes [42] but the initial number of follicles and the yield were not specified. Given the fetuses ages, there were probably ovogonial follicles able to develop. Primordial follicles came later, after 30 weeks of gestation. More recently, Xiao’s, McLaughlin’s and Xu’s teams obtained metaphase II oocytes from adult pubescent human ovarian tissue. The first two cultured COCs on 2D systems (low attachment plate and track-etched nucleopore membranes, respectively) then carried out a final in vitro maturation that lasted 24 to 42 h (Table 5). The yield was quite close to that reported by Zhang et al.: 20% for Xiao and 28% for McLaughlin. Xu’s team did not use a COC culture system, they were grown individually. 21.4% of COCs contained a mature oocyte [25].

In 2015, Xiao’s team obtained the first human metaphase II oocytes from preantral follicles. Sixty-five preantral follicles were mechanically isolated of which 32 evolved to the antral follicle stage. After 40 days, four metaphase II oocytes were obtained (6.1% of follicles became mature oocytes). The main challenges were the long duration and the control of the dynamic environment [23].

In 2018, McLaughlin’s team achieved the first complete in vitro folliculogenesis; metaphase II oocytes were obtained from reserve follicles of pubescent human ovarian tissue. First, fragments of ovarian tissue only containing primordial and/or primary follicles were isolated and cultured for 8 days to reach the secondary follicle stage (160 ovarian tissues cultured and 87 secondary follicles isolated). In vitro secondary follicle culture without a culture system lasted 8 days until the antral follicle stage; it gave 54 antral follicles. Then, among the COCs isolated from these follicles, 32 were matured in vitro; 9 metaphase II oocytes were obtained with spindle and polar body but were dysmorphic. The yield from secondary follicles to metaphase II oocytes was 10.3% but that from primordial follicles to metaphase II oocytes was much lower. Due to ethical constraints, metaphase II oocytes were not fertilized so their developmental competence could not be checked; this multi-step model requires significant experience and expertise; it lasted 21 days [24].

Shortly after, Xu’s team reported complete folliculogenesis using a multi-step model. The first step carried out in tissue allowed primordial follicles to become secondary follicles without antrum. A next follicular isolation step cultured 38 follicles for 6 weeks. The follicle survival rate was 50%. Fourteen COCs were obtained and matured in vitro resulting in 3 mature oocytes in metaphase II. The yield was 7.9% from isolated secondary follicles. The oocytes had similar morphology to in vivo matured oocytes. For the same ethical reasons, developmental competence could not be checked. The total duration was 63 days [25].

## 7. Discussion

The results were interesting and promising; however, there was still room for improvement, particularly regarding the multiple steps used to move from one development stage to another. The use of ovarian tissue to achieve complete in vitro folliculogenesis may be feasible. In 2014, after a 6-week culture, Laronda’s team reported the presence of an antral follicle in the ovarian tissue of an 18-year-old woman without isolation of the follicle at the secondary stage [7]; however, this result has not been reproduced; it would also be interesting to evaluate the pressures required by the follicle according to the stages of development in order to optimize the size of the cultured tissue and the elasticity of the biomaterial that encapsulates the follicles. Another important point is the origin of the tissue. In fact, the results may be different according to whether or not the tissue comes from a pubescent person and whether or not this person has received hormonal treatments (such as androgens in transgender people). In most articles, the tissue is derived from a pubescent human because non-pubescent human tissues are likely to react differently. Long-term treatments of human tissues with androgens would not cause tissue or follicle damage [88] but, in 2019, Telfer reported slower growth of follicles from transgender people but that these follicles still allow obtaining metaphase II oocytes (unpublished data, IVF Worldwide webinar, 2019). In addition, one should note that in vitro culture times are much shorter than in vivo. In the literature, (i) the mean ± SD time of an in vitro first phase of follicular development is 14.8 ± 11.2 days, the progression of follicles from the primordial to the secondary stage is extremely rapid vs. the human physiological delay of about 300 days [22]; (ii) in vitro, the second culture stage lasted 20.5 ± 14.9 days vs. 65 days in vivo [22]; (iii) the growth of antral follicles followed by in vitro maturation of the COCs lasts 10 ± 11.8 days in vitro vs. 20 days in vivo [22]; thus metaphase II oocytes can be obtained after 21 days [49]; these rapid in vitro progressions have not yet proven to produce a developmentally competent oocyte and might not be favorable for obtaining ‘quality cells’. In fact, oocytes obtained in vitro have smaller diameters vs; those obtained in vivo [23,24,38]. 

The longest culture times were reached with 3D culture systems; however complete folliculogenesis has been reported without a culture system. A combination of the results of the 3D culture system and the results of complete folliculogenesis would optimize and potentiate in vitro folliculogenesis. Opting for the most convenient method to perform in vitro folliculogenesis was not easy because there are still several uncertainties about a non-negligible number of culture conditions. Indeed, most studies were carried out with heterogeneous methodologies and small sample sizes; in addition, multi-step techniques require significant expertise and control.

## 8. Conclusions

Reviews of in vitro folliculogenesis results do exist but, to our current knowledge, there are still no reviews on the methods used and the difficulties encountered in the lab work. We believe this review is very useful for research teams who wish to know the present ‘state of the art’ or wish to start research work in that domain.

According to the current procedures, the first phase of in vitro folliculogenesis is carried out within the original ovarian tissue. The addition of an initiation activator did not seem essential but allowed obtaining better yields. The use of a 3D culture system at this stage provided no benefit. The second stage requires a mechanical or/and enzymatic isolation of the secondary follicles and, here too, the use of an activator and/or a 3D culture system were not found necessary. Overall, the results are promising but far from producing several mature and good-quality oocytes whose functionality still has to be checked. For the moment, reaching pregnancies and live births in large animals are essential to validate the current in vitro folliculogenesis techniques.

The success of in vitro folliculogenesis in humans will allow significant advances in understanding ovarian physiology and interesting clinical applications. The technique will be beneficial for patients with premature ovarian failure or for cancer patients at risk of ovarian metastases who have no other solution to recover fertility. The technique is also needed for toxicological assessments of many molecules described as endocrine disruptors; their biochemical mechanisms and effects are still poorly known and are worth being explored in greater depth. 

## Figures and Tables

**Figure 1 biomedicines-10-02217-f001:**
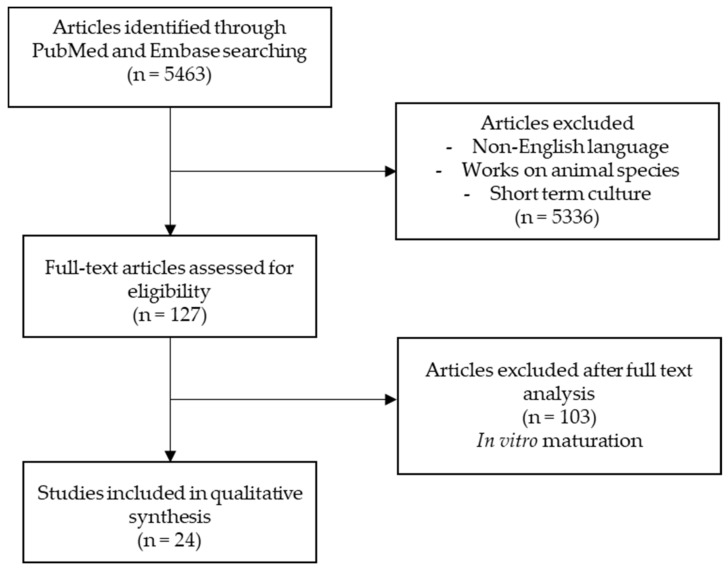
Flow chart of the included studies and search strategy.

**Figure 2 biomedicines-10-02217-f002:**
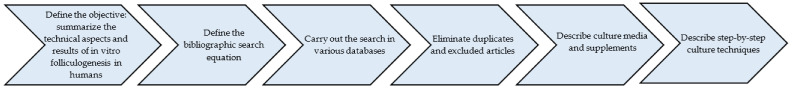
Summary of the steps used to build the review.

**Table 1 biomedicines-10-02217-t001:** Summary of the different culture media used.

Supplements	Range of Doses	Mean ± SD	Role	Specific notes	References
FSH (initial stage)	0.01–1 IU/mL	0.306 ± 0.360 IU/mL	Decreases follicular atresia [41]	Surprising use at a gonadotropin-independent stage	[7,24,27,31,32,33,42,43,44,48,49]
FSH (secondary stage)	0.01–1.36 IU/mL	0.556 ± 0.717 IU/mL	Follicular growth and maturation	In the studies showing the longest culture times and using oocytes in metaphase II, the mean ± SD FSH concentration was 0.017 ± 0.009 IU/mL [23,24,25,42].	[23,28,30,34,39,46,47]
LH (secondary stage, antral stage)	0.03–20 IU/mL	5.05 ± 9.96 IU/mL	Follicular growth and maturation	Only one team added LH right from the beginning of the in vitro culture at a concentration of 0.3 IU/mL [27]	[24,42,45,49]
Insulin	0.01–5000 µg/mL	457.39 ± 1506.61 µg/mL	Action on follicular growth, steroidogenesis, and oocyte maturation [50]	Reduced follicular atresia and stimulated in vitro growth [41,51] Should be controlled; at high concentrations, insulin led to decrease in pre-antral follicle vitality, whereas, at low concentrations, it allowed the same expression of oocyte genes as in fresh tissue. In vitro, insulin reduced follicular atresia and stimulated growth [41,51]	[7,23,24,25,29,30,31,32,33,35,36,37,38,39,46,47]
Selenium	4–6.7 ng/mL	5.04 ± 1.04 ng/mL	Anti-oxidant that reduces the oxidative stress during in vitro culture		[7,23,24,25,29,30,31,32,33,35,36,37,38,46,47]
Glutamine	2–3 mM	2.75 ± 0.5 mM	Essential amino-acid		[24,35,36,37,39,44]
Transferrin	2.5–5500 µg/mL	1053.42 ± 2214.92 µg/mL	Binds to iron to limit its cellular toxicity	One group used concentrations between 2.5 and 6.25 µg/mL [7,23,24,25,29,30,31,32,35,37,38,39,46], and a second used 1000-fold concentrations, 2.5 to 5.5 mg/mL [33,36,47]. At a concentration in the µg range the results were more encouraging; they showed longer growing times and higher follicle growth rates [7,23,24,25,29,30,35,39].	[7,23,24,25,29,30,31,32,33,35,36,37,38,39,46,47]
Ascorbic acid	0.05–50 mg/mL	14.32 ± 24.37 mg/mL	Anti-oxidant that increases follicular vitality (goats [40]; bovine [52]; equine [53])	Concentrations ranging from 50 µg/mL [7,24,34,35,37,39] to 50 mg/mL [32,47].	[7,24,32,34,35,37,39,47,48]
Glucose	NA	NA		Present in AlphaMEM medium (5 mM) and McCoy’s 5a medium (17 mM)	
Sodium pyruvate	0.47 mM	0.47 ± 0.00 mM			[44,47]
8-Bromo-cGMP	2.5 mM	NA			[29]
Bovine fetuin	0.005–1 mg/mL	0.33 ± 0.57 mg/mL			[7,23,33]
Linoleic acid		5.35 µg/mL [46]			

NA: not analyzed.

**Table 2 biomedicines-10-02217-t002:** Studies reporting human in vitro folliculogenesis from primordial to secondary follicle stage.

Authors	Year	Tissue Size (mm)			Culture Period (Days)	Matrices	Use of a Culture System	System Description	Relative Change in the Number of Follicles between Beginning and End of Culture (%)				Peak Day of Secondary Follicles	Observations
		L	W	T					Primordial Follicles	Primary Follicles	Secondary Follicles	Antral Follicles		
Hovatta et al	1997	NA	NA	0.1–0.3	21	Tissue	2-D	Millicel insert + extracellular matrix	+1%	−17%	+14%	+1%	D 21	-
Zhang et al.	1995	1	1	1	25	Tissue	2-D	Costar membrane insert	NA	NA	+49%	0%	NA	-
Wright et al.	1999	NA	NA	1–3	15	Tissue	2-D	Matrigel	−36%	+36% (single category)		0%	D 10	-
Hovatta et al.	1999	NA	NA	0.1–0.3	28	Tissue	2-D	Millicel insert + extracellular matrix	−57%	+31%	+25%	0%	D 7–9	-
Hreinsson et al.	2002	1	1	1	14	Tissue	2-D	Millicel insert + extracellular matrix	−45.9%	−9.4%	+55.3%	0%	D 14	-
Sadeu et al.	2008	1	1	1	28	Tissue	No	-	−33.2%	+26.3%	+6.8%	0%	D 28	Results in a single patient
Telfer et al.	2008	0.5	0.5	0.5	10	Tissue	No	-	−30%	+15%	+15%	0%	D 6	Follicle isolation for continued culture
Lerer-Serfaty et al.	2013	NA	NA	1–2	12	Tissue	3-D	Alginate	−41%	+46% (single category)		0%	D 7	-
Khosravi et al.	2013	2	2	0.5	7	Tissue	No	-	−35.4%	+4.8%	+26.2%	0%	D 7	-
Laronda et al.	2014	1	1	0.5	45	Tissue	3-D	Alginate	NA	NA	NA	1 from 15 ovarian tissues	NA	-
McLaughlin et al.	2014	4	2	1	6	Tissue	No	-	−46.2%	+60% (single category)		0%	D 6	-
Asadi et al.	2017	4	2	1	6	Tissue	No	-	−48.1%	+7.8%	+35.7%	0%	D 6	-
Grosbois et al.	2018	4	2	1	6	Tissue	No	-	−52%	+29%	+35%	0%	D 6	-
McLaughlin et al.	2018	1	1	0.5	8	Tissue	No	-	−37.4%	+29.5%	+7.9%	0%	D 8	Follicle isolation for continued culture
Wang et al.	2014	-	-	-	8	Isolated follicles	3-D	Alginate	−30.8%	−28.5%	+0.8%	+60%	D 8	Follicle isolation for continued culture
Hosseini et al.	2017	-	-	-	10	Isolated follicles	3-D	Alginate	NA	NA	NA	0%	D 10	Report on an increase in follicular diameter (52.4 to 176.4 µm), no count per histological category
Xu et al.	2021	Disc: volume 0.014 mm^3^–diameter 300 µm			21	Tissue	No	-	NA	NA	NA	NA	D 21	No results on the evolution of the follicles but on AMH secretion

D: day; NA: Not Analyzed; 2-D: 2-dimensional; 3-D: 3-dimensional. L: length, W: width, T: thickness.

**Table 3 biomedicines-10-02217-t003:** Studies reporting the use of activators in the first step of follicular growth, from the primary follicle to the secondary follicle.

Authors	Year	Activator	Concentration	Duration of Activation (Days)	Total Culture Duration (Days)	Matrix	Culture System	Relative Change in the Number of Follicles between Beginning and End of Culture (%)			
								Primordial Follicles	Primary Follicles	Secondary Follicles	Antral Follicles
Zhang et al.	1995	No	-	-	25	Tissue	Costar membrane insert	NA	NA	49%	0%
Hovatta et al.	1997	No			21	Tissue	Millicel insert + extracellular matrix	+1%	−17%	+14%	+1%
Wright et al.	1999	No	-	-	15	Tissue	Matrigel	−36%	36% (single category)		0%
Hovatta et al.	1999	No	-	-	28	Tissue	Millicel insert + extracellular matrix	−57%	+31%	+25%	0%
Hreinsson et al.	2002	GDF9	200 ng/mL	14	14	Tissue	Millicel insert + extracellular matrix	−45.9%	-9.4%	+55.3%	0%
Sadeu and Smitz	2008	No	-	-	28	Tissue	No	−33.2%	+26.3%	+6.8%	0%
Telfer et al.	2008	No	-	-	10	Tissue	No	−30%	+15%	+15%	0%
Lerer-Serfaty et al.	2013	bpV/740YP	-	-	12	Tissue	Alginate	−41%	46% (single category)		0%
Khosravi et al.	2013	No	-	-	7	Tissue	No	−35.4%	+4.8%	+26.2%	0%
Laronda et al.	2014	No	-	-	45	Tissue	Alginate	NA	NA	NA	1 out of 15 ovarian tissues
McLaughlin et al.	2014	bpV	1 µM	6	6	Tissue	No	−46.2%	60% (single category)		0%
Asadi et al.	2017	VEGFA165	100 ng/mL	6	6	Tissue	No	−48.1%	+7.8%	+35.7%	0%
Grosbois et al.	2018	mTOR	-	1–2	6	Tissue	No	−52%	+29%	+35%	0%
McLaughlin et al.	2018	No	-	-	8	Tissue	No	−37.4%	+29.5%	+7.9%	0%
Wang et al.	2014	bFGF	200 ng/mL	8	8	Isolated follicles	Alginate	−30.8%	−28.5%	+0.8%	+60%
Hosseini et al.	2017	+/−PRP	-	-	10	Isolated follicles	Alginate	NA	NA	NA	0%
Xu et al.	2021	No	-	-	21	Tissue	No	NA	NA	NA	NA
Chiti et al.	2022	No	-	-	7	Isolated follicles	Alginate + extracellular matrix hydrogel	NA	NA	NA	0%

NA: Not Analyzed. bpV: dipotassium bisperoxo(5-hydroxypyridine-2carboxyl)oxovanadate (V); mTOR: target of rapamycin; VEGF A165: vascular endothelial growth factor A165; PRP: platelet-rich plasma; bFGF: basic fibroblast growth factor; GDF9: growth differentiation factor 9.4.3.3. The PI3K pathway.

**Table 4 biomedicines-10-02217-t004:** Studies reporting isolation of secondary follicle in order to carry out the second step of folliculogenesis (from secondary follicle to antral follicle stage).

Authors	Year	Type of Digestion	Mechanical Digestion	Enzymatic Digestion						Outcomes *
			Needle Diameter	Collagenase	DNase	Neutral Red Solution	Liberase	Time (min)	Temperature (°C)	
Roy et al.	1993	Enzymatic	-	2.4 IU/mL	180 IU/mL	-	-	60	37	Tertiary
Zhang et al.	1995	NM		-	-	-	-	-	-	MII
Abir et al.	1997	Mechanical	21 G	-	-	-	-	-	-	Secondary
Xu et al.	2009	Enzymatic + Mechanical	25 G	0.20%	0.02%	-	-	90	37	Tertiary
Xiao et al.	2015	Mechanical	25 G	-	-	-	-	-	-	MII
Xia et al.	2015	NM	-	-	-	-	-	-	-	Secondary
Yin et al.	2016	Enzymatic	-	0.2 mg/mL	0.2 mg/mL	50 mg/mL	0.04 mg/mL	NM	NM	Tertiary
McLaughlin et al.	2018	Mechanical	25 G	-	-	-	-	-	-	MII
Xu et al.	2021	Mechanical	NM							MII

NA: Not Analyzed. *: final stage obtained. MII: metaphase II oocytes.

**Table 5 biomedicines-10-02217-t005:** Studies reporting human in vitro folliculogenesis from the secondary to the antral follicle stage (step 2) and then from the cumulus-oocyte complex to the metaphase II oocyte (step 3).

Authors	Year	Multistep	Step	Culture System	Biomaterial	Concentration	Duration (Days)	Matrix	Activator	Outcomes					
										Initial Stage	Diameter (µm)	Final Stage	Survival Rate (%)	Diameter (µm)	Start of Antrum
Roy et al.	1993	No	2	2-D	Agar	0.60%	5	Isolated follicles	No	Secondary	NA	Tertiary	NA	NA	NA
Zhang et al.	1995	Yes	2	2-D	Costar membrane insert (collagen)	NA	30–40	Isolated follicles	No	Secondary	60	Tertiary	NA	NA	NA
Zhang et al.	1995	Yes	3	No	-	-	1.5	COC	NA	COC	NA	MII	25%	80	-
Abir et al.	1997	No	2	2-D	Millicell insert	-	28	Isolated follicles	No	Secondary	NA	Secondary	NA	351 ± 270	NA
Abir et al.	1999	No	2	3-D	Collagen	NA	1		No	Secondary	NA	Secondary	0%		NA
Xu et al.	2009	No	2	3-D	Matrigel	33%	30	Isolated follicles	No	Secondary	170.8 ± 51.1	Tertiary	75%	715 ± 68	D 12
Xu et al.	2009		2	3-D	Alginate	0.50%	30	Isolated follicles	No	Secondary	178.4 ± 69.2	Tertiary	75%	715 ± 68	D 12
Krotz et al.	2010	No	3	3-D	3-D granulosa and theca cells	NA	7	COC	No	COC	NA	MII	NA	NA	NA
Xiao et al.	2015	Yes	2	3-D	Alginate	0.50%	10–15	Isolated follicles	No	Secondary	165.8 ± 32.3	Tertiary	NA	500	NA
Xiao et al.	2015	Yes	3	No	-	-	25–30	Isolated follicles	NA	Tertiary	NA	MII *	NA	NA	-
Xia et al.	2015	No	2	3-D	Alginate + mesenchymal stem cells	1%	8	Isolated follicles	No	Secondary	54.7 ± 2.7	Secondary	NA	82.9	NA
Xia et al.	2015	No	2	3-D	Alginate	1%	8	Isolated follicles	No	Secondary	53 ± 3.6	Secondary	NA	69.5 ± 5.5	NA
Yin et al.	2016	No	2	3-D	Alginate	0.30–0.50%	30	Isolated follicles	No	Secondary	184 ± 35	Tertiary	60%	661 ± 120	D 20
McLaughlin et al.	2018	Yes	2	No		-	8	Isolated follicles	Activin A	Secondary	100–150	Tertiary	NA	NA	NA
McLaughlin et al.	2018	Yes	3	2-D	Track-etched nucleopore membranes	NA	4	COC	Activin A	COC	100	MII *	NA	NA	-
Xu et al.	2021	Yes	2	No	-	-	42	Isolated follicles	AMH modulation	Secondary	125.0–198.4	Tertiary	50%	>600	D 21
Xu et al.	2021	Yes	3	No	-	-	1.5	COC	No	COC	NA	MII	NA	>110	-

NA: Not Analyzed. 2-D: two-dimensional culture system; 3-D: three-dimensional culture system; COC: cumulus-oocyte complex; MII: metaphase II oocyte. *: addition of an in vitro maturation step of the cumulus-oocyte complex.

## Data Availability

The data that support the findings of this study are available from the corresponding author upon reasonable request. Author’s roles: Participation in design (E.L., B.S., J.L.), execution and manuscript drafting (E.L.) and critical discussion (E.L., B.S., J.L.).

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
