# Peer review of "An Update on In Vitro Folliculogenesis: A New Technique for Post-Cancer Fertility"

_biomedicines, 2022, doi:10.3390/biomedicines10092217_

Round 1

Reviewer 1 Report

Dear Authors,

The Review titled "Systematic Review An update on in vitro folliculogenesis: a new technique for post-cancer fertility”, is well written and covers a niche area of development.

 A few comments for your notice.

1.       The compiled data from both databases could have been represented in a figure format.

2.       The section on “Description of culture media used for in vitro folliculogenesis” is a cumbersome read. It could have easily been captured in a table.

3.       Table 1 needs to be landscape mode to make it easy to read.

Author Response

Dear Authors,

The Review titled "Systematic Review An update on in vitro folliculogenesis: a new technique for post-cancer fertility” is well written and covers a niche area of development.

A few comments for your notice.

  • The compiled data from both databases could have been represented in a figure format.

The flow chart that summarizes the data as extracted from the databases has been entirely reshaped.

  • The section on “Description of culture media used for in vitro folliculogenesis” is a cumbersome read. It could have easily been captured in a table.

We do agree with this suggestion. The major part of that section is now captured in a new table.

  • Table 1 needs to be landscape mode to make it easy to read.

For an unknown technical reason, all tables initially given to the submission system in landscape mode have been displayed in a portrait mode! Anyway, this table was reshaped (columns merged, etc.) for a simpler and clearer presentation. The other tables of the article underwent also similar changes. All are now presented in a landscape mode.

Reviewer 2 Report

Dear Authors,

I've read with great interest your paper. However, I have some concerns that can be found below;

1) Reference style is not the one mentioned in the Instructions for Authors.

2) Paragraphs are extremely long and can be tiring for the reader.

3) Why use only PubMed and Embase for search?

4) Please create separate Inclusion and Exclusion criteria

5) Why the searches were performed until March 2021? This was more than a year ago.

6) Only 23 articles included? This is an extremely low number

7) Declaration below the flow chart should be deleted and mentioned in the text with the appropriate reference

8) The sections in the tables are too compact. Please find another way to present data.

9) A schematic figure to summarize all info is mandatory.

10) What novelty this systematic review provides to the already existing evidence?

Kind regards and all the best,

The Reviewer

Author Response

Dear Authors,

I've read with great interest your paper. However, I have some concerns that can be found below.

  • Reference style is not the one mentioned in the Instructions for Authors.

The references are now revised and put in the journal’s style.

  • Paragraphs are extremely long and can be tiring for the reader.

We do agree with this comment. One of the most bulky paragraphs of the manuscript (section 3) has been transformed into a table; the second one (section 4) has been significantly shortened.

  • Why use only PubMed and Embase for search?

The manuscript reports only on the most successful searches in the most popular and/or reliable databases. In fact, searches made in other databases (Web of science, Google Scholar, etc.) returned either duplicates or articles far from the scope or inclusion criteria of this literature review. This is now added in Methods.

  • Please create separate Inclusion and Exclusion criteria

The paragraph dedicated to these criteria has been fully revised to display more clearly the inclusion and exclusion criteria.

  • Why the searches were performed until March 2021? This was more than a year ago.

One reason for that date is that the data displayed in the tables required a lot of time for detailed analyses and renewed checks and that the manuscript was not rapidly written.

Anyway, we carried out the same search strategy over an additional year (until end of March 2022). This led to find only one article whose content is now added to previous results and tables.

  • Only 23 articles included? This is an extremely low number.

This number is indeed low. We have been probably too restrictive regarding the language (English only) and the species (human only). Nevertheless, the technique is relatively recent and there are currently very few teams dedicated to the search in that difficult domain. This is now added to Methods.

  • Declaration below the flow chart should be deleted and mentioned in the text with the appropriate reference

We do agree with this comment. In fact, a reference is not (or is no more) required for this flow chart because it has been entirely revised.

  • The sections in the tables are too compact. Please find another way to present data.

In the revised manuscript, all the tables are now reshaped (columns merged, etc.) for a simpler and clearer presentation.

  • A schematic figure to summarize all info is mandatory.

We have drawn a new figure that summarizes the steps of the approach used to build this review. We hope this will give a useful overview.

  • What novelty this systematic review provides to the already existing evidence?

Reviews of in vitro folliculogenesis results do exist but, to our current knowledge, there are still no reviews on the methods used and the difficulties encountered in the lab work. We believe this review is very useful for research teams who wish to know the present ‘state of the art’ or wish to start research work in that domain. This is now added to the Conclusion.

Kind regards and all the best,

The Reviewer

Thank you very much for your kind attention,

The authors

Round 2

Reviewer 2 Report

Dear Authors,

You did an excellent job. Based on the revised version, I think it is suitable for publication.